# Recombinant Virus Quantification Using Single-Cell Droplet Digital PCR: A Method for Infectious Titer Quantification

**DOI:** 10.3390/v15051060

**Published:** 2023-04-26

**Authors:** Ksenija Korotkaja, Anna Zajakina

**Affiliations:** Cancer Gene Therapy Group, Latvian Biomedical Research and Study Centre, Ratsupites Str. 1, k.1, LV-1067 Riga, Latvia; ksenija.korotkaja@biomed.lu.lv

**Keywords:** alphaviruses, Semliki Forest virus, replication-deficient virus particles, ddPCR, infectious titer, single-cell ddPCR

## Abstract

The quantification of viruses is necessary for both research and clinical applications. The methods available for RNA virus quantification possess several drawbacks, including sensitivity to inhibitors and the necessity of a standard curve generation. The main purpose of this study was to develop and validate a method for the quantification of recombinant, replication-deficient Semliki Forest virus (SFV) vectors using droplet digital PCR (ddPCR). This technique demonstrated stability and reproducibility using various sets of primers that targeted inserted transgenes, as well as the nsP1 and nsP4 genes of the SFV genome. Furthermore, the genome titers in the mixture of two types of replication-deficient recombinant virus particles were successfully measured after optimizing the annealing/extension temperature and virus:virus ratios. To measure the infectious units, we developed a single-cell ddPCR, adding the whole infected cells to the droplet PCR mixture. Cell distribution in the droplets was investigated, and β-actin primers were used to normalize the quantification. As a result, the number of infected cells and the virus infectious units were quantified. Potentially, the proposed single-cell ddPCR approach could be used to quantify infected cells for clinical applications.

## 1. Introduction

The quantification of viruses is necessary for both research and clinical application. Quantitative real-time PCR (qPCR) is frequently used to determine the genome titer of viruses. The genome titer can be measured in genome equivalents per mL (ge/mL), viral genomes per mL (vg/mL), or viral particles per mL (vp/mL), assuming that each viral particle contains one copy of the genome [1]. However, the qPCR must be coupled with a standard curve and is sensitive to PCR inhibitors [2]. Moreover, for the quantification of RNA viruses, first, complementary DNA (cDNA) should be synthesized. Additionally, the genome titer does not allow for a precise estimation of the virus’s capacity for infection (infectious units, iu) and, therefore, cannot be utilized to assess the multiplicity of infection (MOI) during experiments. Other methods, such as plaque assay, which results in the quantification of plaque forming units (pfu) or TCID50 (tissue culture infectious dose 50%), can be used to quantify the titer of lytic viruses [3,4]. These methods involve measuring the ability of the virus to infect and kill cells, which provides a measure of viral infectivity. However, they may not be suitable for quantifying the titer of non-lytic viruses or replication-deficient viral vectors. The immunostaining of infected cells could be used to quantify the infectious titer. However, specific antibodies for modified viral vectors are not usually offered commercially.

Droplet digital PCR (ddPCR) is a third-generation PCR in which the reaction can be divided into approximately 20,000 water-in-oil droplets [5,6]. The presence or absence of amplification in each droplet can be measured using fluorescence-based detection, allowing the absolute quantification of target nucleic acid in the sample. By counting the number of positive and negative droplets, ddPCR can provide the absolute quantification of the number of viral particles in a sample. As a result of the partitioning of the reaction, this technique is less sensitive to PCR inhibitors and does not require a standard validation curve. This method allows the quantification of RNA in a one-step RT-ddPCR reaction by combining cDNA synthesis and ddPCR in one mixture.

Recently, it has been demonstrated that ddPCR can be employed to detect salmonid alphavirus from seawater [7] and accurately quantify adeno-associated viral vectors [8,9]. During SARS-CoV-2 RNA detection in wastewater with ddPCR, Semliki Forest virus (SFV) vectors were employed as a standard to evaluate the effectiveness and success rate of SARS-CoV-2 extraction techniques [10]. Single-cell ddPCR has previously been used to analyze the ratio of genetically modified cells after stem cell gene therapy and to detect male fetal cells in maternal blood samples [11,12]. We assumed that this method could potentially be used for the direct detection of viruses in cells and the quantification of infected cells in tissues.

The main purpose of this study was to develop and validate a method for the quantification of alphaviral vectors using ddPCR. Alphaviruses are enveloped (+) ssRNA viruses belonging to the Togaviridae family that have been proposed as potential gene therapy delivery systems [13,14]. Alphaviruses are promising vectors for cancer immunotherapy due to their broad tissue tropism, low pre-immunity, and efficient transgene expression [15,16,17]. Furthermore, alphaviruses can be used as a test system for antiviral research due to their close resemblance to other disease-causing viruses [18,19]. In this study, two recombinant replication-deficient Semliki Forest virus vectors that encode fluorescent protein genes (CFP or DS-Red) were employed as model virus particles since their infectious titers can be easily determined using the fluorescent microscopy of infected cells. A gradual increase in virus particle concentration was used to optimize genome titer quantification. The genome titers in the mixture of two types of recombinant virus particles, expressing either CFP or DS-Red genes, were measured by varying the annealing/extension temperature and virus:virus ratio (the ratio of the number of SFV/DS-Red recombinant virus particles to the number of SFV/CFP recombinant virus particles within the sample). To quantify the infectious units, we employed single-cell ddPCR. The results of this study indicate that ddPCR could be used to measure both the genome and the infectious titers of Semliki Forest virus vectors.

## 2. Materials and Methods

### 2.1. Cell Lines 

Baby hamster kidney fibroblasts (BHK-21) were obtained from the American Type Culture Collection (Cat. No. CCL-10™; ATCC/LGC Prochem, Boras, Sweden) and cultured in Glasgow’s MEM (Cat. No. 21710025; Gibco, Life Technologies, Thermo Fisher Scientific, Waltham, MA, USA) supplemented with 5% Fetal Bovine Serum (FBS; Cat. No. FBS-HI-12A; Capricorn Scientific, Ebsdorfergrund, Hessen, Germany), 10% Tryptose Phosphate Broth solution (Cat. No. 18050039; Gibco, Life Technologies), 20 mM HEPES (Cat. No. 15630056; Gibco, Life Technologies), 2 mM L-Glutamine (Cat. No. 25030024; Gibco, Life Technologies), 50 U/mL penicillin, and 50 mg/mL streptomycin (Cat. No. 15070063; Gibco, Life Technologies). Cells were incubated in a humidified 5% CO_2_ incubator at 37 °C and passaged using a 0.05% trypsin solution (Cat. No. 15400054; Gibco, Life Technologies).

### 2.2. Cell Staining and Treatment

BHK-21 cells were stained with WGA Alexa Fluor 488 (Cat. No. W11261; Invitrogen, Waltham, MA, USA) following the manufacturer’s instructions. Briefly, the cells were incubated with 1 μg/mL of WGA solution in Phosphate-Buffered Saline (PBS) at 37 °C. After 10 min, the cells were washed and used for the experiments. 

### 2.3. Plasmids 

The pSFV1-DsRed and pSFV1-CFP vectors encoding, accordingly, the *Discosoma* sp. red fluorescent protein (DS-Red) and cyan fluorescent protein (CFP) genes were generated in our lab as described previously [20]. Prof. Henrik Garoff (Karolinska Institute, Sweden) generously provided the pSFV1 and pSFV-Helper1 plasmids [21].

### 2.4. Production of Replication-Deficient Viral Particles

Replication-deficient viral particles were produced as described previously and illustrated in Figure 1 [22]. The restriction enzyme SpeI (BcuI; Cat. No. ER1251; Thermo Fisher Scientific, Waltham, MA, USA) was used to linearize the plasmids pSFV1-DsRed, pSFV1-CFP, and pSFV-Helper1. The linearized plasmids were purified by using PrepEase^®^ DNA Clean-Up Kit (Cat. No. 78759; USB, Cleveland, OH, USA). Additionally, 1 µg of each plasmid was used as a template for in vitro transcription with SP6 RNA polymerase (Cat. No. AM2071; Thermo Fisher Scientific, USA) according to the manufacturer’s instruction in a total volume of 50 µL. The transcribed RNAs were capped by adding a 1 mM 3′-O-Me-m7G(5′)ppp(5′)G cap-structure analog (Cat. No. S1411S; New England Biolabs, Ipswich, MA, USA). Then, the DNA template was removed with DNase I (Cat. No. EN0523; Thermo Fisher Scientific, USA). To package the RNAs into viral particles, each of the recombinant RNAs was co-electroporated with the SFV-Helper1 RNA in 1 × 10^7^ BHK-21 cells by pulsing the mixture twice at 850 V, 25 mF using a Gene Pulser-II system (Bio-Rad, Hercules, CA, USA). The electroporated BHK-21 cells were resuspended in 12 mL of a BHK-21 medium supplemented with 5% FBS and incubated at 33 °C in an atmosphere containing 5% CO_2_ in a humidified incubator. After 48 h, the cell cultivation medium containing the infectious viral particles was harvested, centrifuged (10,000× *g*, 20 min, 4 °C, Eppendorf fixed-angle rotor 5804), filtered through a 0.22 µm pore vacuum filter (Cat. No. 83.1822.001, Sarstedt, Nümbrecht, Germany), and purified/concentrated by ultracentrifugation through double sucrose cushions (50% and 20% sucrose in TNE buffer) as previously described [23]. Briefly, 30 mL of a filtered virus particle-containing medium was carefully overlayed onto a sucrose two-step gradient and centrifuged using an SW32Ti rotor (Beckman Coulter, Brea, CA, USA) at 150,000 × *g* for 90 min at +4 °C. Virus particle-containing fractions (2 mL) were collected from the bottoms of the pierced tubes and dialyzed for 5 h against a TNE buffer using dialysis cassettes (Cat. No. 02906-36; Spectrum Labs, Rancho Dominquez, CA, USA). The concentrated viral particles were aliquoted, frozen in liquid nitrogen, and stored at −70 °C.

### 2.5. Infection

BHK-21 cells were cultivated to an 80–90% monolayer density. Cells were washed twice with PBS Ca^2+^/Mg^2+^, and then the viral particle solution was diluted with PBS Ca^2+^/Mg^2+^, which was added to the cells. The cells were incubated at 37 °C for 1 h. After incubation, a viral particle-containing solution was removed from each well, and the medium supplemented with 1% FBS was added. Infected cells were incubated overnight in a humidified 5% CO_2_ incubator at 37 °C.

### 2.6. Infectious Virus Particle Titer Determination

Infectious virus particle titers were determined by fluorescent microscopy as previously described [24]. Briefly, BHK-21 cells were infected with serial dilutions of viral particles. After 24 h, the infected cells expressing DS-Red and CFP were counted in 10 view fields using fluorescent microscopy (Leica DM-IL). The SFV viral particle titers were expressed as infectious units per mL (iu/mL) and were used later to calculate the multiplicity of infection (MOI).

### 2.7. Primer Design

Primers and probes were designed using Primer3Plus software (Whitehead Institute for Biomedical Research, Massachusetts Institute of Technology) as the BioRad ddPCR guide recommended. Probes were designed with 6-Fam or Hex fluorophores at the 5′ end and BHQ-1 quencher at the 3′ end. All primers were synthesized by Metabion (Germany) and PCR tested at T_a_ = 60 °C. The primer/probe set was named in correspondence to its starting base pair in the plasmid. The CFP_8011 primer set was described previously by Gudra et al. [10]. The β-actin primer set was modified from Moreira et al. [25]. The primers and probes used in this study are shown in Table 1.

### 2.8. Reverse Transcriptase Droplet Digital PCR (RT-ddPCR)

The RT-ddPCR was performed using a One-Step RT-ddPCR Advanced Kit for Probes (Cat. No. 1864021; Bio-Rad, USA) according to the manufacturer’s instructions. Briefly, the 20 μL reaction mixture was composed of 5 μL Supermix, 2 μL reverse transcriptase, 1 μL of DTT, 900 nM primers, 250 nM probes, a virus particle containing a sample or the infected cell sample (without an RNA extraction step), and diethylpyrocarbonate (DEPC)-treated H_2_O (Cat. No. R0601; Thermo Scientific, USA). Droplets were generated using a QX200 Droplet Generator (Bio-Rad, USA), and PCR was performed with C1000 or T100 Thermal Cycler (Bio-Rad, USA). Cycling conditions were 1 h at 50 °C, 10 min at 95 °C, followed by 40 cycles of a two-step thermal profile composed of 30 s at 95 °C and 60 s at 60 °C or 61.4 °C at a ramp rate of 1 °C/s before 10 min at 98 °C. To acquire a higher accepted droplet number, ddPCR plates were incubated at 12 °C for at least 4 h after cycling, as recommended by Rowlands et al. [26]. After cycling, the plate was transferred to a QX200 Droplet Reader (Bio-Rad, USA). Data analysis was performed using QuantaSoft software (version 1.7.4.0917; Bio-Rad, USA). The contamination with plasmid DNA was controlled by standard PCR without an RT-step and using respective primer sets.

### 2.9. Single-Cell Droplet Digital PCR

The infected cells were collected by trypsinization 24 h after infection. Next, cells were filtered through 30 μm pre-separation filters (Cat. No. 130-041-407; Miltenyi Biotec, Bergisch Gladbach, Germany), washed with PBS, centrifuged, and resuspended in PBS supplemented with 10% FBS (10% FBS). Then, the cells were counted and stored on ice until their addition to the one-step RT-ddPCR reaction mix. The optimal cell number for ddPCR is 1000–2000 cells in 4 μL 10% FBS. It is recommended to generate droplets within 5 min after adding the cells to the reaction mix. Droplet generation with a QX200 Droplet Generator was carried out as described above. Briefly, the PCR mixture was transferred to the DG8 Droplet Generator Cartridge (Cat. No. 1864008; Bio-Rad, USA). Then, 70 µL of Droplet Generation Oil for Probes (Cat. No. 1863005; Bio-Rad, USA) was added to the relevant wells in the cartridge, which was then covered with a DG8 Gasket (Cat. No. 1863009; Bio-Rad, USA) and placed into the Droplet Generator. After droplet generation, the encapsulation of the cells into droplets could be tested using microscopy. PCR and further droplet analysis were performed as described above.

### 2.10. Statistical Analysis

The data were expressed as the mean of at least three experiments ± the standard error of the mean (SEM). Statistical analysis was performed using GraphPad Prism 8.02 software.

## 3. Results

### 3.1. Quantification of SFV Virus Particles by ddPCR and Its Correlation with Infectious Units

First, the ddPCR-counted genome titer’s correlation with the infectious titer of the recombinant virus particles was explored. Two recombinant replication-deficient alphaviruses were used for the experiments—SFV/DS-Red and SFV/CFP—encoding red and cyan fluorescent genes, respectively. Replication-deficient alphaviruses with inserted fluorescent protein genes were used because their titers (iu/mL) could be easily calculated using the fluorescent microscopy of infected cells. Before RNA packaging into viral particles, the DNA template was removed using DNase I as DNA contamination could affect the ddPCR quantification. Furthermore, virus particles were purified by ultracentrifugation on a double sucrose cushion as described by the methods to remove non-encapsulated RNA. The infectious titer of the virus particles was quantified using direct fluorescent microscopy of infected cells. The primers and probes were designed using Primer3Plus software. ddPCR was performed using a one-step RT-ddPCR kit without viral RNA extraction.

The quantification of the genome titer was optimized using a virus particle concentration gradient (Figure 2a,b). ddPCR plots with a set of primers/probes targeting transgenes (DS-Red_7816, CFP_8011) showed an efficient positive and negative droplet separation. The genome titer that resulted in the best positive and negative droplet separation was considered to be the optimal titer for SFV ddPCR quantification, providing the most efficient performance of positive and negative droplet populations with a minimal presence of “rain” (droplets that fall between the major positive and negative populations). The optimal genome titer for SFV/DS-Red ddPCR quantification was found to be ~(0.4–1.4) × 10^7^ virus particles per mL (vp/mL), which is equal to (2.2–6.7) × 10^5^ SFV/DS-Red infectious units per mL (iu/mL). The optimal genome titer for SFV/CFP ddPCR quantification was found to be ~8.1 × 10^6^ vp/mL, which is equal to 2.2 × 10^6^ SFV/CFP iu/mL.

The quantified genome titer correlated with an infectious titer of both types of virus particles (Figure 2c; $R_{SFV/DS–Red}^{2}=0.99997$; $R_{SFV/CFP}^{2}=0.9998$). However, the results showed that the relation between the ddPCR quantified genome titer and infectious titer was individual for each virus particle preparation. For example, ${\left(\frac{Genometiter}{Infectioustiter}\right)}_{SFV/DS–Red}=20.16\pm 0.05,$ while ${\left(\frac{Genometiter}{Infectioustiter}\right)}_{SFV/CFP}=3.53\pm 0.02$. This difference could be explained by virus particle instability and the presence of varying amounts of non-infectious or damaged particles and non-incapsulated RNAs. Due to these limitations, the ddPCR of viral suspension could not be used to quantify infectious titer.

### 3.2. Different Primers Show Similar Results

Various primer sets were used to investigate whether the quantified genome titer was affected by the primer set. Probes against DS-Red, nsP1, and nsP4 genes were designed with two different fluorophores—FAM and HEX—to investigate whether the targeted genes or the fluorophore itself affected quantification. As a result, calculated genome titers obtained using different sets of primers were in the same range (~2.8 × 10^8^ vp/mL, Figure 3). The method demonstrates stability and reproducibility using different sets of primers to target the transgene or the SFV genome, respectively.

### 3.3. Quantification of a Mixture of Two Types of Virus Particles: SFV/DS-Red and SFV/CFP

The capacity to measure more than one target in the same sample is one of the primary benefits of ddPCR. For various reasons, for example, in cancer gene therapy, viral vectors expressing different therapeutic genes can be used in combination [27,28,29]. Therefore, it was investigated whether the amount of two types of recombinant virus particles could be quantified in one sample.

Virus particles expressing DS-Red and CFP genes have been mixed according to the calculated optimal genome titer for ddPCR quantification (the resulting $\frac{SFV/DS–Red}{SFV/CFP}$ proportion calculated by infectious titers is $\frac{1}{10}$). Next, the genome titers of the premixed SFV/DS-Red and SFV/CFP virus particles were quantified with DS-Red_7816-FAM and CFP_8011-HEX probe sets. First, to achieve the best cluster separation, the thermal gradient was performed by adjusting the temperature of annealing/extension (T_a_) from 65 to 55 °C. ddPCR plots show how lowering T_a_ could improve the separation of negative and positive droplets (Figure 4a,b). However, a lower T_a_ may increase the primers’ non-specific binding; therefore, the highest temperature, which produced the most effective cluster separation (T_a_ = 61.4 °C), was used in further experiments (Figure 4c). Double-positive droplets are the result of SFV/DS-Red and SFV/CFP viral particles being randomly encapsulated into one droplet.

In order to measure the two premixed types of virus particles, different SFV/DS-Red:SFV/CFP ratios were examined. This method showed stability; while the genome titer of the first alphavirus remained constant, the genome titer of the second alphavirus showed linear regression (Figure 5a). If the SFV/CFP was used at a constant concentration and SFV/DS-Red was in a range from 2.7 × 10^3^ to 2.0 × 10^6^ iu/mL, R^2^_SFV/DS-Red_ = 0.9997, similarly, in the case that SFV/DS-Red was applied at a constant concentration and SFV/CFP was in a range from 2.7 × 10^4^ to 2.0 × 10^7^ iu/mL, R^2^_SFV/CFP_ = 0.9997.

Furthermore, dilution-adjusted genome titers were similar in both cases (Figure 5b). If SFV/CFP was at a constant concentration and SFV/DS-Red varied, the calculated genome titers were (3.4 ± 0.3) × 10^8^ vp/mL for SFV/DS-Red and (1.68 ± 0.04) × 10^9^ vp/mL for SFV/CFP. On the other hand, if SFV/DS-Red was at a constant concentration and SFV/CFP varied, the calculated genome titers were (3.16 ± 0.10) × 10^8^ vp/mL for SFV/DS-Red and (1.62 ± 0.03) × 10^9^ vp/mL for SFV/CFP, respectively. Therefore, the calculated virus particle genome titers were close in both approaches. Additionally, the titers calculated for SFV/DS-Red in the mix of two types of virus particles were very close to the titers obtained in the single SFV/DS-Red quantification (~2.8 × 10^8^ vp/mL, Figure 3). The possibility of quantifying various virus particles is an important advantage of ddPCR in clinical sample analysis.

Next, we investigated the quantification of premixed replication-deficient recombinant alphaviruses with significantly different titers. For this purpose, two SFV/DS-Red:SFV/CFP virus particle ratios were compared: $\frac{SFV/DSRed}{SFV/CFP}=\frac{1}{7}$ and $\frac{SFV/DSRed}{SFV/CFP}=\frac{1}{70}$. Virus particle concentrations were adjusted using infectious titers and were calculated using fluorescent microscopy (iu/mL). The correlation between the ddPCR quantified genome titer and infectious titer was quantified for both mixes. The variability increased significantly in the 1:70 sample compared to the 1:7 sample, with the coefficient of determination decreasing from 0.9991 to 0.94 in the SFV/DS-Red case and from 0.998 to 0.986 in the SFV/CFP case (Figure 6a,b). In the 1:70 virus particle mix, titers of SFV/DS-Red and SFV/CFP were (5 ± 3) × 10^8^ vp/mL and (1.3 ± 0.4) × 10^9^ vp/mL, respectively, while in the 1:7 mix, they were (5.1 ± 0.8) × 10^8^ vp/mL and (1.48 ± 0.11) × 10^9^ vp/mL, respectively (Figure 6c). Thus, the virus:virus ratio is an important parameter for accurate virus particle quantification.

### 3.4. Infected Cell ddPCR

To address the problem of infectious titer quantification, the whole cell ddPCR for the quantification of the ssRNA virus particle titer was developed. To calculate the number of infectious particles in mL, the proportion of infected cells has to be quantified. The schematic step representation of infected cell ddPCR for quantification of virus particle infection titers is shown in Figure 7.

Firstly, fluorescently labeled cell encapsulation and distribution in the droplets were investigated. Uninfected BHK-21 cells were collected using trypsin and filtered through 30 μm pre-separation filters; then, cells were resuspended in PBS supplemented with 10% FBS (10% FBS) and counted. A total of 20,000 BHK-21 cells were stained with WGA Alexa Fluor 488 and added to the one-step ddPCR mix. Droplets were generated and studied under a fluorescent microscope (Figure 8a,b). Cell distribution in the droplets was not homogenous, with 60.4% empty droplets, 28.3% droplets containing one cell, 8.5% droplets containing two cells, and 2.8% containing three cells. The experiment shows the need to normalize cell numbers using an internal standard.

β-actin primers were generated to detect cell encapsulation. A 10% FBS solution was tested to confirm the absence of a non-specific amplification in ddPCR (Appendix A). The non-specific signal was not detected with either single β-actin-FAM or duplex β-actin-FAM and DS-Red-HEX primer sets.

To measure the number of cells, the cells were collected and used for ddPCR quantification with β-actin-FAM. Igarashi et al. suggested that using 2000 cells per reaction mix provided optimal quantification [11]. Thus, 1000 and 2000 cells were added to the reaction mix. Droplet generation and PCR were carried out as described above for the quantification of virus particle-containing solutions (Figure 8c). The same experiment was performed with a duplex primer set: β-actin-FAM and DS-Red-HEX. Although the DS-Red-HEX signal of uninfected cells was negative as expected, the duplex primers decreased the intensity shift between positive and negative droplets in the FAM channel (Figure 8c. see arrows). The less effective droplet separation could potentially decrease the sensitivity of the method. Remarkably, the number of cells quantified by ddPCR using β-actin primers varied significantly between the experiments and could be different from two to ten times compared to the “real” cell number added to the reaction mix. This can be explained by cell degradation during the trypsinization and droplet formation, resulting in the appearance of free-floating β-actin-RNA, which was detected by sensitive PCR. This finding is consistent with previous reports of a similar effect [11]. Nevertheless, we assumed that the relationship between infected and uninfected cells would not be significantly affected by cell degradation in the same experiment. Therefore, the quantification of viral RNA by whole-cell ddPCR was carried out.

The cells were infected with SFV/DS-Red and analyzed using a β-actin-FAM and DS-Red-HEX primer combination. Additionally, it was tested whether cell permeabilization was needed for whole cell ddPCR as this is usually utilized for in situ RT-PCR [30]. Cell treatment with 0.02% Triton X-100 decreased the amplification signal (Appendix A) due to RNA degradation, as shown previously [31,32]. It was concluded that cell permeabilization did not improve ddPCR quantification.

The use of fluorescent cells infected with SFV/DS-Red could potentially affect the probe detection signal. Therefore, we tested whether the cell fluorescence generated a non-specific signal (Appendix A). For this experiment, a high MOI (MOI = 11) was used, which resulted in the bright DS-Red expression in BHK-21 cells. ddPCR was performed with and without β-actin-FAM and DS-Red-HEX primers. As a result, no unspecific signal was detected in the sample without primers; therefore, it was concluded that DS-Red protein fluorescence did not interfere with the ddPCR signal detection. We supposed that DS-Red was denatured during the PCR.

Finally, to quantify the infectious titer, BHK-21 cells were infected with SFV/DS-Red at a different MOI. The initial (reference) infectious titer, calculated by fluorescent microscopy, was (1.73 ± 0.11) × 10^7^ iu/mL. As a result of ddPCR, the high MOI provided an increase in the double-positive β-actin^+^SFV/DS-Red^+^ droplet fraction (Figure 9a). For the quantification of the infectious virus particles, the proportion of infected cells had to be multiplied by the total cell number to obtain the number of infectious units (iu). To calculate the titer (iu/mL), the number of infectious units (iu) should be multiplied by virus particle sample dilution and divided by the volume of the virus particles used for infection (mL).

First, for infectious titer quantification, the proportion of infected cells was calculated as the double-positive droplet number divided by the number of β-actin positive droplets. The following equation was used:(1)$${Titer}_{SFV/DS–Red}=\frac{{(\beta –actin}^{+}{DS–Red}^{+})\times N_{cells}\times {Dilution}_{SFV/DS–Red}}{\left({\beta –actin}^{+}\right)\times V_{SFV/DS–Red}}$$
where *β-actin^+^DS-Red^+^* is the number of double-positive droplets;

*N_cells_* is the total number of cells used for infection;*Dilution_SFV/DS-Red_* is the dilution of the virus particles used for infection;*β-actin^+^* is the number of β-actin positive droplets;*V_SFV/DS-Red_* is the volume of SFV/DS-Red used for infection (mL).

Free-floating RNAs were discovered due to cell disruption and damage or an excess of viral particles. Therefore, some of the double-positive droplets may have been non-infected cells co-encapsulated with the free-floating RNA. The linkage value (copies/μL) proposed by Bio-Rad is a measure of the degree of association between two variables or factors. It is calculated as the number of double positives over and beyond those predicted by random distribution. In this case, linkage could be used to measure the β-actin-FAM and DS-Red-HEX signal from the same cell. The QuantaSoft Analysis Pro calculated linkage was correlated with MOI and, therefore, suitable to describe the percentage of infected cells (Figure 9b). This value was applied to calculate the proportion of infected cells by dividing it by the QuantaSoft Analysis Pro calculated concentration of β-actin (copies/μL). For the quantification of the infectious virus particle titer using linkage, the following equation was used:(2)$${Titer}_{SFV/DS–Red}=\frac{Linkage\times N_{cells}\times {Dilution}_{SFV/DS–Red}}{c_{\beta –actin}\times V_{SFV/DS–Red}}$$
where *Linkage* is the number of double positives over and beyond those predicted by random distribution (copies/μL);

*N_cells_* is the total number of cells used for infection;*Dilution_SFV/DS-Red_* is the dilution of the virus particles used for infection;*c_β-actin_* is the QuantaSoft Analysis Pro quantified concentration of β-actin (copies/μL);*V_SFV/DS-Red_* is the volume of SFV/DS-Red used for infection (mL).

The application of these two approaches *(1)* using *double-positive droplet numbers* or *(2) Linkage* produced different infectious titers (Figure 9c). The infectious titers calculated using *double-positive droplet numbers* were higher—(1.1 ± 0.5) × 10^8^ iu/mL, and differed significantly from the initial titer (1.73 ± 0.11) × 10^7^ iu/mL), calculated by the direct fluorescent microscopy analysis of infected cells, which could be explained by randomly distributed free-floating RNAs compromising a part of the double-positive droplet population. On the other hand, *Linkage*-calculated titers—(1.0 ± 0.3) × 10^7^ iu/mL—were very close to the reference titer, making the *Linkage* approach more accurate and promising. The major disadvantage of this approach was that *Linkage* could not be calculated if the droplet separation was not proper; for example, in Figure 9a, MOI = 0.074.

For successful quantification, an uninfected cell fraction (β-actin^+^DS-Red^−^) was required using both ddPCR-based approaches. β-actin^+^DS-Red^−^ was necessary to adequately calculate the proportion of infected cells. Therefore, for ddPCR quantification, it was essential to find the optimal MOI. Although the same problem occurs using the immunostaining approach, immunostaining provides more variability as the titer can be counted not only at a different MOI but also at different microscope magnifications.

## 4. Discussion

In this study, we presented ddPCR as a method to quantify the number of viral particles and the virus particle infectious titer. The accurate quantification of virus particle titers is crucial for various research applications, such as virus-based gene therapy, vaccine development, and antiviral drug discovery, as well as for clinical diagnostics. Researchers continue to develop and improve methods for measuring virus titers and virus loads with greater precision and sensitivity [33,34,35]. The titer of the non-lytic virus is typically quantified using qPCR, but this method has several limitations as it can only measure the amount of viral nucleic acid and cannot distinguish between infectious and non-infectious virus particles. It has been demonstrated previously and proven in this study that the vector genome titer and infectious genome titer are significantly different [1,36]. Immunostaining is a useful method for determining infectious titers, but it is a time-consuming and challenging process requiring specific antibodies. On the other hand, a primer design for qPCR or ddPCR is generally easier and less time-consuming than generating custom antibodies.

Virus particle-mediated therapeutic gene delivery is a promising gene therapy approach, including virus-based vaccines [15,37]. RNA viruses present specific challenges when quantifying their concentration due to the low stability of RNA molecules and the need to convert RNA into cDNA through a reverse transcription step before performing PCR. The low stability of RNA molecules also requires careful handling and storage to prevent degradation and ensure the accurate quantification of viral concentrations. Puglia et al. used RT-PCR to determine the temperature-induced degradation of alphaviral RNA that negatively correlate with the infectivity of the virus [38]. Previous studies have demonstrated that ddPCR can be employed to detect salmonid alphavirus from seawater [7]. However, the accurate quantification of virus titers requires more precise and sensitive methods, as the concentration of viral particles in a sample can vary widely and may be affected by various factors, such as the growth conditions of host cells or the efficacy of the virus purification method.

To improve the quantification of the genome titer, we utilized various sets of primers targeting different regions of the Semliki Forest virus (SFV) genome. Our results showed that genome quantification was both stable and reproducible. Additionally, it was possible to identify the genome concentration in a mixture of two replication-deficient recombinant alphaviruses by adjusting the annealing/extension temperature and the virus:virus ratio. These findings suggest that it is important to use viruses with similar concentrations to optimize the quantification of mixed viruses.

Furthermore, the detection of viral RNA without RNA extraction and a standard curve, as well as quantifying various genes in the same sample, presents a major advantage for ddPCR-based virus particle quantification. Nevertheless, measuring genome equivalents does not provide reliable information on the number of infectious particles in the sample. To quantify the infectious titer, we performed a single-cell ddPCR, which was previously described to analyze various cell subsets [11,12]. We applied two methods for data analysis to quantify the infectious titer. It was demonstrated that the Linkage-based assay provided data very close to the reference titers obtained by fluorescent microscopy, whereas the direct calculation of the double-positive population (β-actin^+^/DS-Red^+^) was less suitable. The use of the housekeeping gene (β-actin) is crucial for accurate quantification, allowing both infected and uninfected cells to be distinguished. However, the effect of free-floating RNAs and the encapsulation of more than one cell or virus particle into one droplet is unclear.

The cell-based ddPCR technique does not allow cell numbers to be adequately quantified using β-actin primers, as well as the genome titers and infectious titers cannot be linked. A second housekeeping gene or another virus gene may be introduced in the reaction, allowing the application of a Linkage-based quantification of a double-positive population. For example, we hypothesized that using two sets of primers against different regions of the viral genome would provide a more accurate quantification, which is close to the infectious titer.

A single-cell ddPCR could be applied to diagnostics with high sensitivity [39,40]. Our data provide insights into novel clinical applications of this approach for the detection of infected cells in clinical samples. Potentially, this can be used to detect HIV-infected cells without DNA extraction. A similar technique was previously suggested by Yucha et al. [41]. The ddPCR approach has already been validated for the quantification of HIV-1 in DNA extracted from PBMC [42]. A single-cell ddPCR can be used not only for virus detection but also as an alternative to immunostaining. For example, single-cell ddPCR can potentially be applied for liver tissue analysis to quantify HBV and HCV-infected hepatocytes in patient biopsies [43].

Overall, ddPCR is a useful method with which to quantify the number of viral particles and the infectious titer. Nevertheless, this method possesses some setbacks and limitations; therefore, standard approaches, e.g., immunostaining, and plaque assay, continue to be useful. The advantages and potential limitations of the ddPCR for virus particle quantification are summarized in Table 2.

It is important to note that ddPCR may not be suitable for all types of virus particles and that additional validation studies are required to ensure the accuracy and reliability of the method.

## 5. Conclusions

A method for alphavirus quantification using ddPCR was developed. The data obtained in this study indicated that ddPCR could be used to quantify both the genome and the infectious titers of the Semliki Forest virus. The developed single-cell ddPCR method could potentially be suitable for the infectious titer quantification of non-lytic or replication-deficient virus particles using the Linkage-based calculation of the housekeeping gene (β-actin) and the viral gene in infected cells. We also concluded that adjusting the annealing/extension temperature and optimizing the virus:virus ratio was crucial for the accurate quantification of a mixture of two types of virus particles.

## Figures and Tables

**Figure 1 viruses-15-01060-f001:**
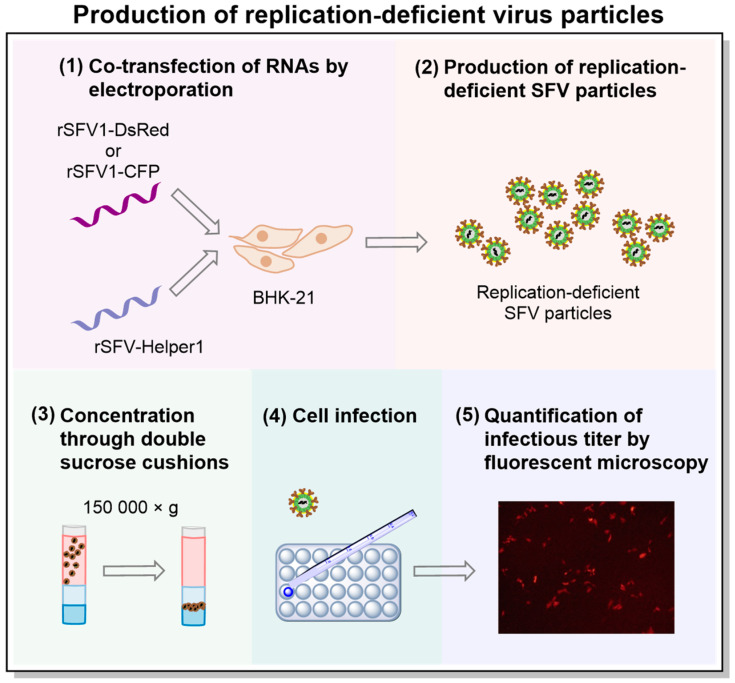
The scheme illustrating the process of preparing Semliki Forest virus (SFV) replication-deficient virus particles. (**1**) The packaging of recombinant RNAs (rSFV1-DsRed or rSFV1-CFP) into virus particles was achieved by co-electroporation with SFV-Helper1 RNA encoding virus structural genes. The helper RNA was not packaged into the particles, ensuring the production of replication-deficient SFV. (**2**) The cell cultivation medium containing replication-deficient virus particles was harvested, centrifuged, and filtered. (**3**) Virus particles were purified/concentrated by ultracentrifugation through double sucrose cushions. (**4**) Cells were infected with SFV particles. (**5**) About 24 h after infection, cells expressing the fluorescent protein gene (DS-Red or CFP) were counted using fluorescent microscopy, and the infectious titer was calculated as infectious units per mL (iu/mL).

**Figure 2 viruses-15-01060-f002:**
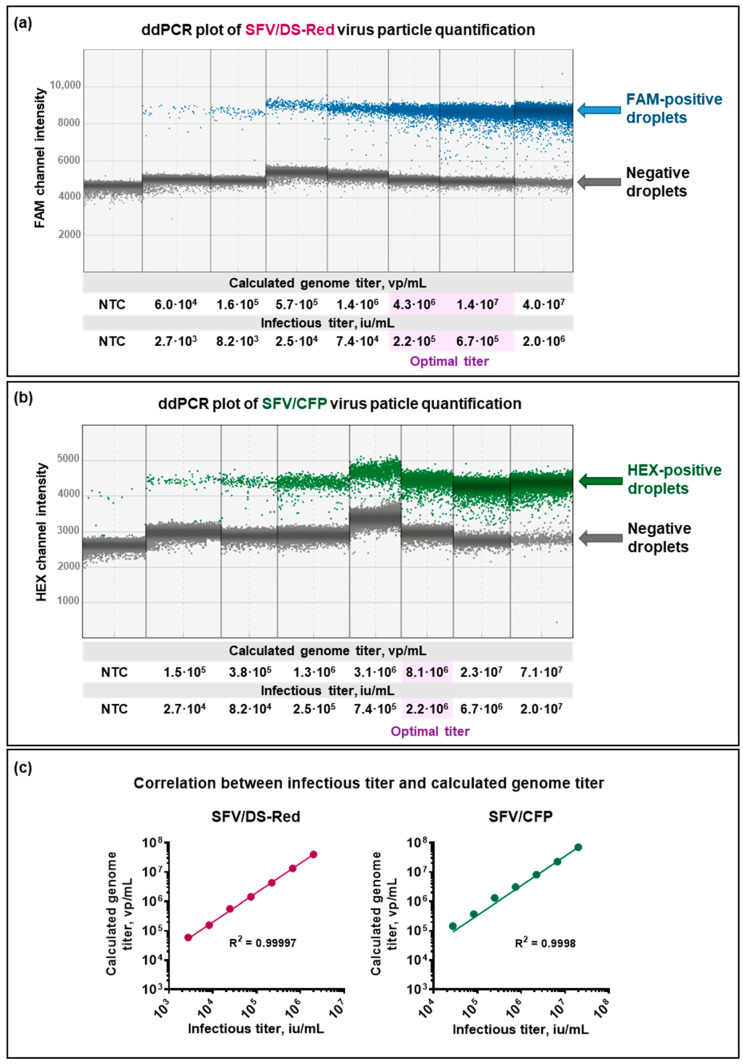
Correlation between virus particle genome equivalents (genome titer, vp/mL) and infectious titer (iu/mL). The recombinant SFV particles (SFV/DS-Red and SFV/CFP) with respective infectious titers were used for the quantification of genome equivalents using ddPCR. The titer of the genome that allowed for the clearest separation between positive and negative droplets was chosen as the optimal titer to quantify SFV particles using ddPCR. (**a**) Quantification of SFV/DS-Red using DS-Red_7816-FAM primer set; (**b**) Quantification of SFV/CFP using CFP_8011-HEX primer set. Each column represents an individual well of ∼20,000 droplets with a respective set of primers/probes. NTC—non-template control; blue—droplets positive in FAM channel; green—droplets positive in HEX channel; purple—optimal titer; grey—negative droplets. (**c**) Correlation between the infectious titer and the genome titer.

**Figure 3 viruses-15-01060-f003:**
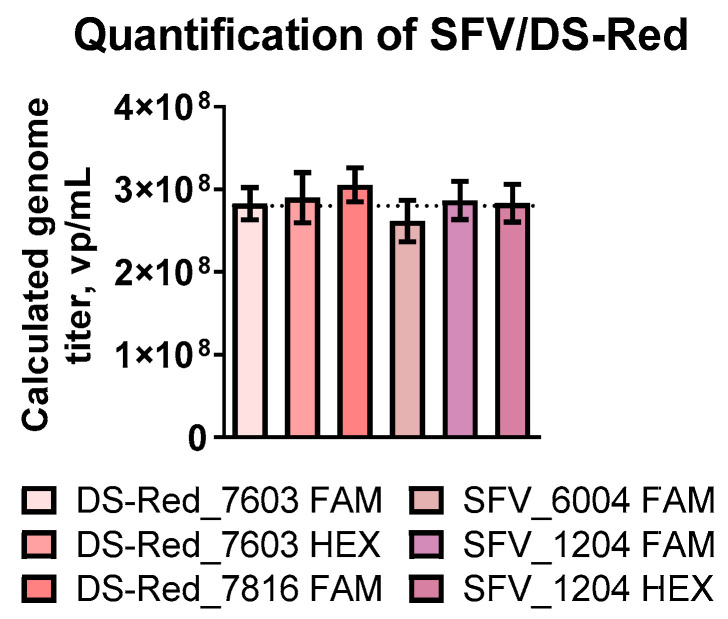
ddPCR quantification of the SFV/DS-Red virus particles with different primer sets. Primer and probe set targeting the DS-Red transgene (DS-Red_7603-FAM, DS-Red_7603-HEX, DS-Red_7816-FAM) and SFV genome (SFV_6004-FAM, SFV_1204-FAM, SFV_1204-HEX) were used for quantification. FAM and HEX indicate the fluorophore used in the probe design. Data are expressed as the mean of at least three experiments ± the standard error of the mean (SEM).

**Figure 4 viruses-15-01060-f004:**
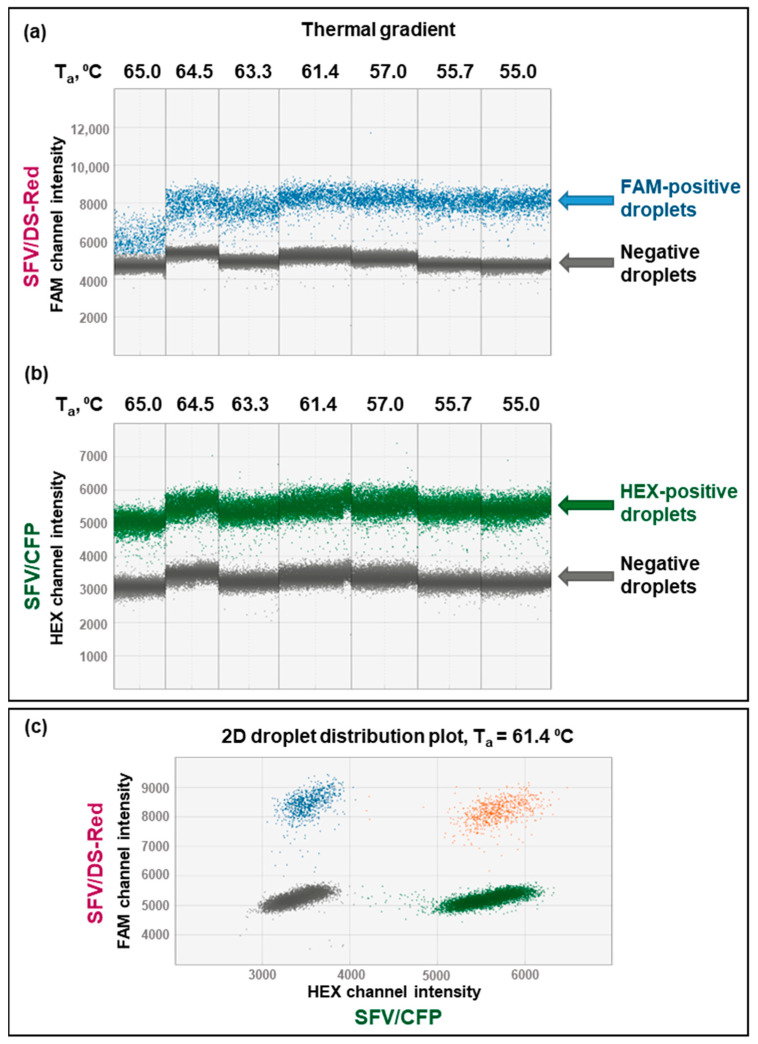
Temperature optimization of premixed SFV/DS-Red and SFV/CFP virus particles $\left(\frac{SFV/DS–Red}{SFV/CFP}=\frac{1}{10}\right)$ amplified with DS-Red_7816-FAM and CFP_8011-HEX primer/probe set combinations. (**a**,**b**) A thermal gradient for duplex SFV/DS-Red and SFV/CFP quantification: (**a**) DS-Red FAM channel, (**b**) CFP HEX channel. (**c**) Two-dimensional (2D) droplet distribution at optimal annealing/extension temperature (T_a_ = 61.4 °C). Blue—droplets positive in the FAM channel; green—droplets positive in the HEX channel; orange—droplets positive in both FAM and HEX channels; grey—droplets negative in both FAM and HEX channels.

**Figure 5 viruses-15-01060-f005:**
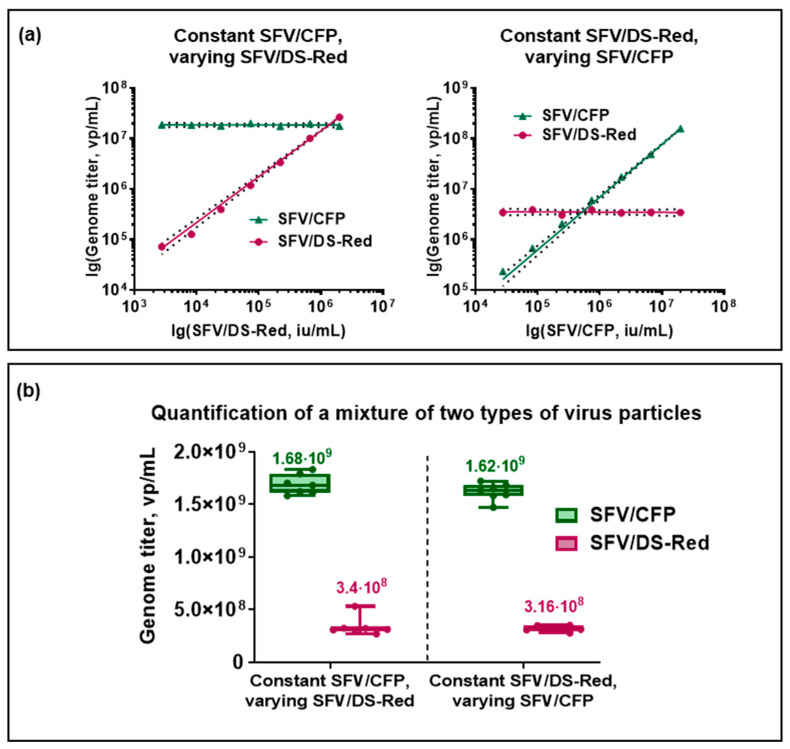
Quantification of premixed SFV/DS-Red and SFV/CFP virus particles with DS-Red_7816-FAM and CFP_8011-HEX primer/probe set combination. The concentration of one alphavirus remained constant while the concentration of the other increased. (**a**) A linear regression between ddPCR determined genome titer (vp/mL) and the number of infectious virus particles added to the reaction mixture (iu/mL). (On the left) SFV/CFP is constant, SFV/DS-Red varies (R^2^ = 0.9997); (on the right) SFV/DS-Red is constant, SFV/CFP varies (R^2^ = 0.9997). The dashed line represents a 95% confidence interval. (**b**) Boxplot graph of virus particle genome titer per mL quantified by ddPCR in the mix of two types of virus particles.

**Figure 6 viruses-15-01060-f006:**
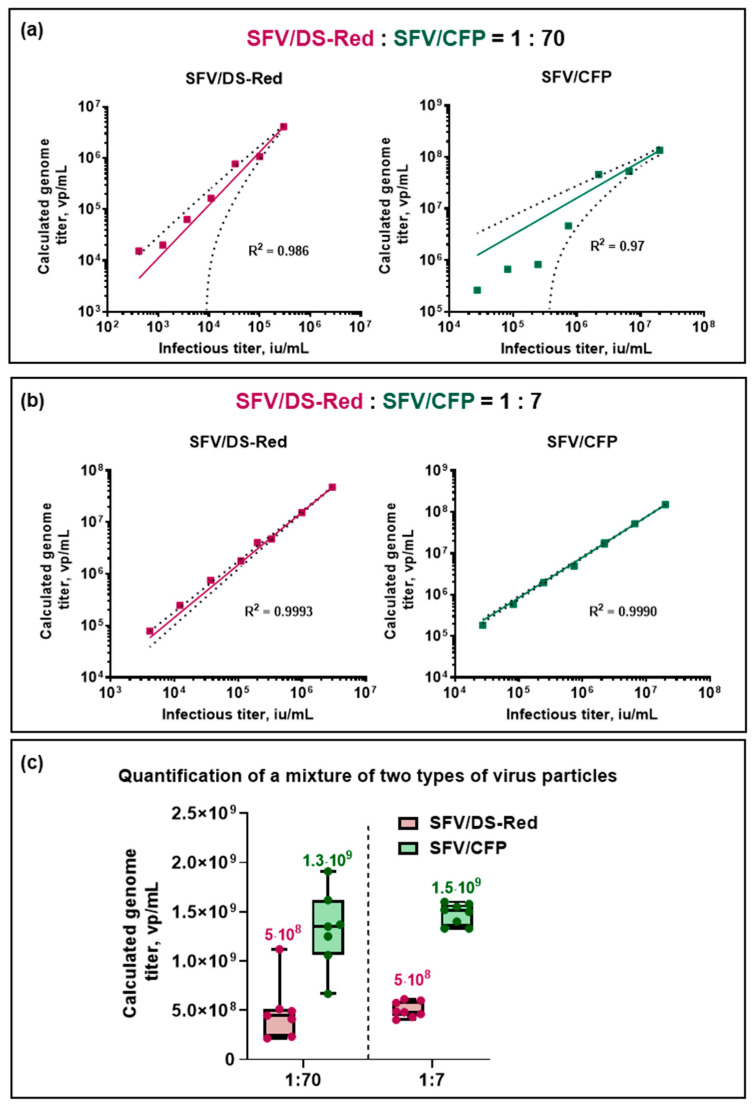
Quantification of premixed SFV/DS-Red and SFV/CFP virus particles with a DS-Red_7816-FAM and CFP_8011-HEX primer set combination. Two types of recombinant virus particles were premixed in two SFV/DS-Red:SFV/CFP ratios: 1-to-70 and 1-to-7. (**a**,**b**) The linear regression between the infectious titer and the ddPCR calculated genome titer: (**a**) $\frac{SFV/DS–Red}{SFV/CFP}$= $\frac{1}{70};$ (**b**) $\frac{SFV/DS–Red}{SFV/CFP}$ = $\frac{1}{7}$. The dashed line represents a 95% confidence interval. (**c**) Boxplot graph of virus particle genome titers quantified in the mix with an SFV/DS-Red:SFV/CFP ratio of 1:70 and 1:7.

**Figure 7 viruses-15-01060-f007:**
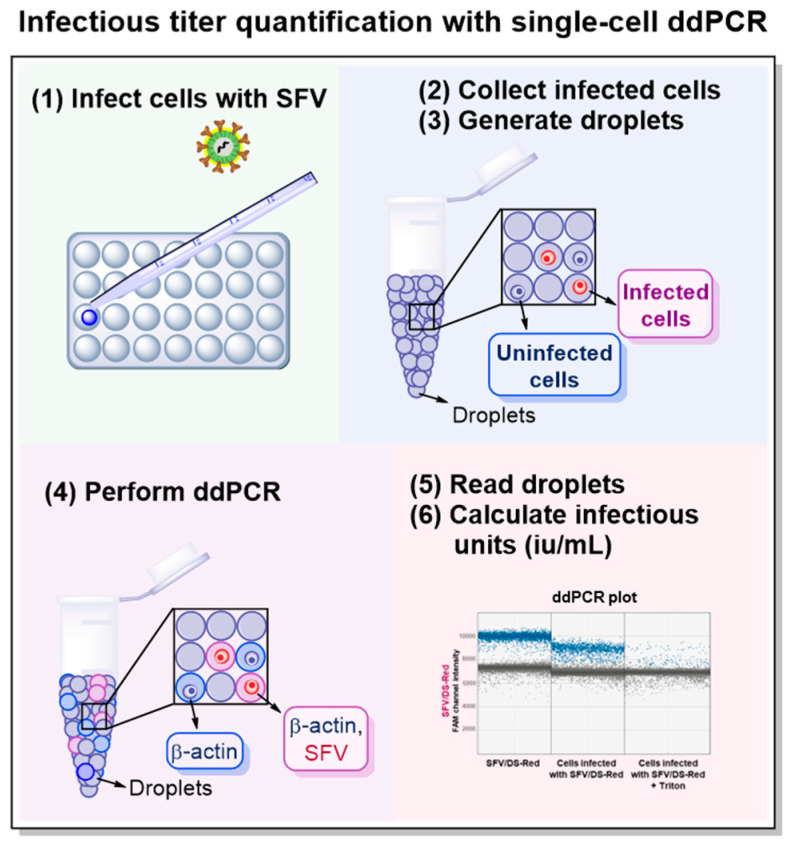
Scheme illustrating the steps involved in the quantification of the infectious virus particle titer with single-cell ddPCR. (**1**) Cell infection with replication-deficient recombinant Semliki Forest virus (SFV). (**2**) Infected cell collection by trypsinization 24 h after infection. After collection, cells should be filtered, washed, and counted. (**3**) Cell encapsulation into droplets with QX200™ Droplet Generator. (**4**) Reverse Transcriptase ddPCR with two primer/probe sets: first targeting the cell housekeeping gene (β-actin) and second targeting viral genome (SFV). (**5**) Droplet primary analysis with QX200™ Droplet Reader and further analysis using droplet distribution ddPCR plots. (**6**) Calculation of infectious titer.

**Figure 8 viruses-15-01060-f008:**
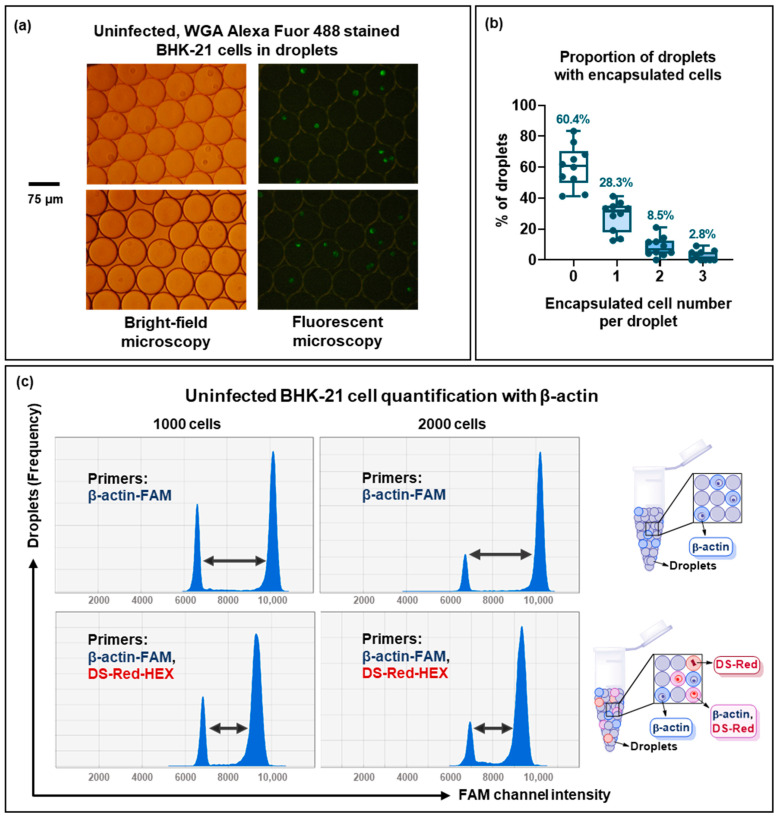
Single-cell ddPCR. (**a**,**b**) Uninfected BHK-21 cells were collected, stained with WGA Alexa Fluor 488 (green), counted, and then 20,000 cells were encapsulated into droplets with Droplet Generator: (**a**) Bright-field and fluorescent microscopy of droplets with uninfected BHK-21 cells stained with *WGA Alexa Fluor 488*; (**b**) Proportion of droplets containing 0, 1, 2, and 3 encapsulated cells. (**c**) Histogram of 1000 and 2000 uninfected BHK-21 cells quantified using *(i)* only β-actin-FAM primers or *(ii)* β-actin-FAM primers in combination with DS-Red_7603-HEX primers. Arrows indicate shifts in fluorescence intensity.

**Figure 9 viruses-15-01060-f009:**
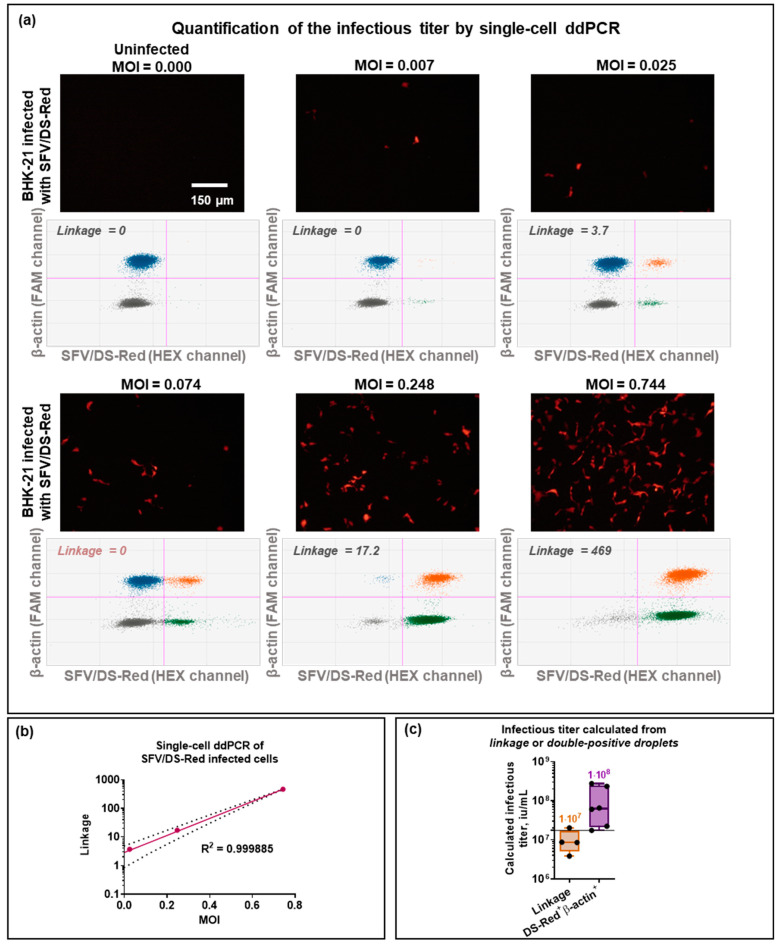
Quantification of SFV/DS-Red infectious titer with single-cell ddPCR. BHK-21 were infected with SFV/DS-Red virus particles, encoding the red fluorescent protein gene. After 24 h, infected cells were photographed, collected, counted, and 2000 cells were added to the ddPCR reaction to quantify β-actin and DS-Red genes with a β-actin-FAM and DS-Red_7603-HEX primer set combination. (**a**) Fluorescent microscopy and 2D droplet distribution of BHK-21 cells infected with SFV/DS-Red at several MOI. Blue—droplets positive in the FAM channel; green—droplets positive in the HEX channel; orange—droplets positive in both FAM and HEX channels; grey—droplets negative in both FAM and HEX channels. (**b**) Correlation between MOI and *Linkage* between β-actin-positive and DS-Red-positive droplet populations calculated in *QuantaSoft Analysis Pro*. The dashed line represents a 95% confidence interval. (**c**) Calculated infectious titer of SFV/DS-Red using *Linkage* and *double-positive droplet number*. The line represents the reference infectious titer (1.73 × 10^7^ iu/mL) determined by direct fluorescent microscopy. MOI—a multiplicity of infections.

**Table 1 viruses-15-01060-t001:** Primer design. pSFV1-DsRed and pSFV1-CFP plasmid map indicating non-structural protein gene (nsP1–4) and transgene locations. The numbered areas on the map indicate regions where amplification occurred. The primer/probe set was named according to the starting base pair of the corresponding plasmid.

Plasmid Map				
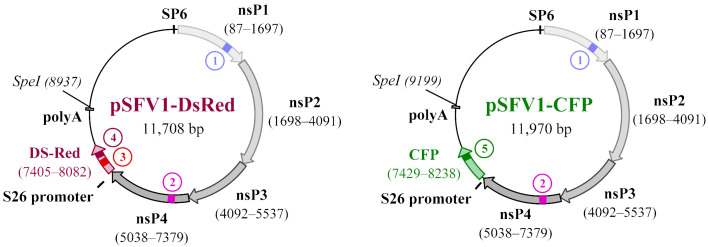				
No	Primer Set	Forward Primer	Probe	Reverse Primer
*Primers and probes targeting the nsP1 gene of the SFV1 vector*				
1.	SFV_1204	5′-CAC AGC GAA ACA CTA ACA CG-3′	5′-6-Fam-CTG CTT CCG ATT GTG GCC GTC-BHQ-1-3′	5′-CAG CAG CAA GTA AGT GAC C-3′
			5′-Hex-CTG CTT CCG ATT GTG GCC GTC-BHQ-1-3′	
*Primers and probes targeting the nsP4 gene of the SFV1 vector*				
2.	SFV_6004	5′-ATA GTT GCT TGG ACA GAG CG-3′	5′-6-Fam-CTA CAG AAC GTG CTA GCG GC-BHQ-1-3′	5′-AGT CCA TGG TGG GTA GTT CT-3′
*Primers and probes targeting the DS-Red gene*				
3.	DS-Red_7603	5′-GCT CCA AGG TGT ACG TGA AG-3′	5′-6-Fam-CCC GCC GAC ATC CCC GAC TAC-BHQ-1-3′	5′-CCT TGT AGA TGA AGG AGC CG-3′
			5′-Hex-CCC GCC GAC ATC CCC GAC TAC-BHQ-1-3′	
4.	DS-Red_7816	5′-AAG AAG ACT ATG GGC TGG G-3′	5′-6-Fam-TAC CCC CGC GAC GGC GTG C-BHQ-1-3′	5′-AGC TTG GAG TCC ACG TAG TAG-3′
*Primers and probes targeting the CFP gene*				
5.	CFP_8011	5′-CTG CTG CCC GAC AAC CAC-3′	5′-Hex-CCA GTC CGC CCT GAG CAA AGA CC-BHQ-1-3′	5′-TCA CGA ACT CCA GCA GGA C-3′
*Primers and probes targeting β-actin*				
6.	β-actin	5′-AGC ACC ATG AAG ATC AAG ATC ATT-3′	5′-6-Fam-CAC TGT CCA CCT TCC AGC AGA-BHQ-1-3′	5′-CGG ACT CAT CGT ACT CCT GCT T-3′

**Table 2 viruses-15-01060-t002:** Advantages and limitations of RNA virus particle quantification with ddPCR.

Advantages	Limitations
Does not need a standard curve	Some droplets may contain more than one virus particle, or cell, which affects quantification
2. Does not require RNA extraction	2. The direct ddPCR of virus particle suspension does not provide information about infectious titer
3. Allows the quantification of genome equivalents of two viruses in one sample; however, the virus:virus ratio should be considered	3. The optimal infected/uninfected cell ratio is required for infected cell quantification
4. Allows the quantification of an infectious titer of recombinant virus particles by infected cell ddPCR 5. Can potentially be applied to infected cell quantification in clinical samples	4. The optimal virus:virus ratio is required for the quantification of a mixture of two types of virus particles 5. Currently not suitable for cell quantification 6. Expensive compared to the standard RT-PCR

## Data Availability

The data presented in this study are available on request from the corresponding author.

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
