# Peer review of "Recombinant Virus Quantification Using Single-Cell Droplet Digital PCR: A Method for Infectious Titer Quantification"

_viruses, 2023, doi:10.3390/v15051060_

Round 1

Reviewer 1 Report

In this manuscript the authors reported the establishment of virus quantification methods using Single-Cell droplet digital PCR using recombinant Semiliki Forest virus. Although, this method needs to be modified since it has a lot of limitations for quantification of virus, it gives the new insight for quantifying virus. The reviewer suggests some minor revisions, before publishing on viruses.

Line 187-191

Although, the authors mentioned about “the optimal genome titer” for ddPCR quantification, it is not clear how “the optimal genome titer” was calculated. Please add the basis of those values.

Line 282-283, Table 2

The authors mentioned that “the virus:virus ratio” is important for accurate virus quantification. Since this method is applied for quantification of virus, it is unlikely that you can adjust the ratio before titration of virus. Since there is such limitation, “Allow quantifying genome equivalents of two viruses in sample” should not be mentioned in advantages of this ddPCR. 

Reviewer 2 Report

The method for alphavirus quantification using ddPCR is presented that can be used to quantify both (1) the genome equivalents and (2) the infectious titers of the Semliki Forest virus (SFV), an alphavirus. Method can be especially useful to titer the non-lytic viruses, which is usually quantitated using qPCR. The latter cannot distinguish between infectious and non-infectious virus particles. Thus, ddPCR can be a useful method for quantifying the number of viral particles and the infectious titer of SFV. Overall it can be a useful method among other methods to quantify viruses and can be a valuable addition to the diagnostic tools.

While the proposed ddPCR method is of interest to virus diagnostic field, some weaknesses need to be addressed. Authors describe titration of recombinant virus. However, they describe infectious titer of non-replicating SFV particles, not a titer of SFV. Terminology needs to be clarified. It would be good to evaluate if ddPCR method is applicable to titer the wild type SFV. While the ddPCR can be applied, other methods can be used for the same purpose. For example, immunostaining method is routinely used for determining infectious titers for non-lytic viruses. For most known viruses, immunostaining reagents are readily available. Additionally, findings suggest that it is important to use viruses with similar concentrations to optimize the ddPCR-based quantification of mixed viruses. This suggests that alternative method such as immunostaining needs to be used in coordination with ddPCR.

Also, please provide additional Figure to illustrate process of preparation of replication-deficient virus particles and indicate location of genes and primers.

Minor comments:

42           Add 1-3 sentences to provide additional general background information on ddPCR methodology.

65           Explain ‘virus:virus ratio’ definition

89           Clarify terminology to explain that replication deficient particles are used in the study. This is different from the virus.

94           Indicate volume of in vitro transcription reaction

163         As described above – please provide additional details.

170         Please specify are these “infectious virus” particles or non-infectious “virus-like” particles.

Table 1  Please provide plasmid map and indicate primer and gene locations.
